# COVID-19 Vaccination and Alcohol Consumption: Justification of Risks

**DOI:** 10.3390/pathogens12020163

**Published:** 2023-01-19

**Authors:** Pavel A. Solopov

**Affiliations:** Frank Reidy Research Center for Bioelectrics, Old Dominion University, Norfolk, VA 23508, USA; psolopov@odu.edu; Tel.: +1-757-683-2416

**Keywords:** COVID-19, vaccines, alcohol, immune system, spike protein, SARS-CoV-2

## Abstract

Since the beginning of the severe acute respiratory syndrome coronavirus 2 (SARS-CoV-2) pandemic, pharmaceutical companies and research institutions have been actively working to develop vaccines, and the mass roll-out of vaccinations against COVID-19 began in January 2021. At the same time, during lockdowns, the consumption of alcoholic beverages increased. During the peak of vaccination, consumption remained at high levels around the world, despite the gradual relaxation of quarantine restrictions. Two of the popular queries on search engines were whether it is safe to drink alcohol after vaccination and whether this will affect the effectiveness of vaccines. Over the past two years, many studies have been published suggesting that excessive drinking not only worsens the course of an acute respiratory distress syndrome caused by the SARS-CoV-2 virus but can also exacerbate post-COVID-19 syndrome. Despite all sorts of online speculation, there is no specific scientific data on alcohol-induced complications after vaccination in the literature. Most of the published vaccine clinical trials do not include groups of patients with a history of alcohol-use disorders. This review analyzed the well-known and new mechanisms of action of COVID-19 vaccines on the immune system and the effects of alcohol and its metabolites on these mechanisms.

## 1. Introduction

Alcohol consumption, especially chronic heavy drinking, has an impact on human health, particularly on the components of both innate and adaptive immunity [1,2]. A large number of early and recent studies have demonstrated that both short- and long-term alcohol consumption leads to a severe decrease in lymphocytes [3,4]. Alterations in immunoglobulins IgA and IgM have been observed in men and women who drink alcohol [5,6]. Ethanol dose- and time-dependently modulates the functions of monocytes and dendritic cells, thereby affecting phagocytosis and inflammatory cytokine production [7]. Interactions between alcohol and the immune system may also influence the development and progression of some types of cancer [8]. It is generally accepted that moderate alcohol consumption, unlike chronic alcohol intoxication, enhances the response to classical vaccines [1]. Tragic events are taking place in the modern world, and they are affecting mental health and increasing global alcohol consumption. However, the same events are stimulating the development of biotechnology and the production of vaccines.

The new coronavirus disease outbreak first identified in China was officially reported on 31 December 2019, and within two weeks, researchers had published the DNA sequence of SARS-CoV-2, the virus that causes COVID-19. By February, the first COVID-19 vaccine candidate (mRNA-1273) had been designed and manufactured by a company called Moderna [9]. Immediately after the lockdown was announced, there was a significant increase in the retail sales of alcohol, with a simultaneous decrease in the retail sales of food, indicating an increase in domestic alcohol consumption during this period [10]. Some countries completely prohibited the sale of liquor, while others reported an increase in activity in the alcohol black market [11,12]. It is known that people who have problems with alcohol experience strong anxiety and drink more to cope with it [13]. Mass methanol poisoning occurred in Iran and led to the deaths of more than 700 people in March 2020 after a rumor circulated in the country that drinking alcohol can prevent being infected by the new virus [14].

In December 2020, the first two COVID-19 vaccines, Pfizer/BioNTech (New York, NY, USA) and Moderna (Cambridge, MA, USA), received the US Food and Drug Administration (FDA) and European Medicines Agency (EMA) Emergency Use Authorization [9]. Mental health researchers reported that the news about the developed vaccines and their “emergency use” status created an additional feeling of anxiety and fear and that people had great doubts about the vaccines’ efficiency and safety [15,16,17,18]. The clinical trials of COVID-19 vaccines, approved in the USA, did not explicitly include individuals with alcohol-use disorders [19]. Our published study indicating that the SARS-CoV-2 spike protein alone may cause acute lung injury in mice caused a great resonance in the press and resulted in questions about whether the new vaccines are safe [20]. We later published a study showing that alcohol exposure exacerbates spike-protein-induced lung damage [21]. The aim of this review article is to determine the link between the immune response to COVID-19 vaccines and the modulation of the immune system by alcohol consumption.

## 2. The Impacts of Alcohol Consumption on the Immune System

Alcohol modulates both innate and adaptive immunity. Several lines of evidence suggest that epithelial cells, macrophages, and dendritic cells, as the first lines of immune defense, are the most susceptible to high doses of alcohol. Ethanol weakens the ability of leucocytes to migrate to sites of infection; induces functional abnormalities in T and B lymphocytes, natural killer cells, and macrophages; and alters cytokine expression [22]. Elevated serum levels of tumor necrosis factor α (TNFα) and interleukin (IL-6), together with decreased IL-10, interferon γ (IFN-γ), and IL-2 levels, are the usual parameters of patients with chronic alcoholic liver disease [23,24].

Many published articles suggest that alcohol consumption has a dose-dependent effect on the response to infection. Those with alcohol disorders are 3–7 times more susceptible to bacterial pneumonia and tuberculosis, and they are prone to the progression of chronic viral infections, such as human immunodeficiency virus (HIV) and hepatitis C [7]. Alcohol metabolism varies from person to person. It depends not only on the sex and constitution of the person but also on the content of metabolizing enzymes in the liver [25]. The liver, an important component of the innate immune system, when damaged due to chronic alcohol abuse, results in the decreased production of antibacterial proteins, thereby increasing the susceptibility to bacterial or viral infection. Patients with alcohol disorders also have an increased susceptibility to respiratory pathogens and an increased risk of acute respiratory distress syndrome (ARDS) [26]. Awaya et al., in their review suggested avoiding alcohol during COVID-19 vaccination [27].

Not only chronic alcohol use can lead to negative effects on the immune system **[2]**. In fact, studies show that heavy drinking also affects the immune system. It has been reported that patients with acute alcohol intoxication are more prone to peritonitis development following penetrating abdominal trauma [28]. In experimental models, acute alcohol intoxication has been demonstrated to impair the mucociliary defense of airways against invading pathogens [29].

However, some animal and clinical studies suggest that moderate alcohol consumption reinforces the immune response to infection and vaccination. In a clinical study that included 391 patients with a cold who were exposed to different respiratory viruses, Co-hen et al., found that consuming a large number of alcoholic drinks (3–4/day) decreased the risk of developing colds that were confirmed by clinical symptoms and specific antibody titers [30]. Some of the health benefits of moderate beer consumption may be due to its ability to interfere with pro-inflammatory cytokine cascades [31].

It should be remembered that the severity of the effects of the immune status in people with alcohol-use disorders depends not only on the time of consumption and the amount of alcohol but also on age, sex, body composition, environmental factors, and even the type of alcoholic beverage [32]. Antioxidants and polyphenols found in red wine and phytoestrogens and vitamins found in beer could be protectors against immune cell damage and cytokine overexpression [33,34,35]. Ethanol can harm immune cells due to the formation of free radicals during metabolism, but antioxidants should provide protection against this [36]. Interestingly, the degree of a hangover the day after alcohol consumption does not affect alcohol-induced immune changes [37].

## 3. “Spike Effect” of COVID-19 Vaccines and Alcohol

The SARS-CoV-2 spike protein (S protein, SP) is a clove-shaped transmembrane structural glycoprotein that is localized on the surface of the SARS-CoV-2 virus [38]. This unit is responsible for the recognition of and the binding to the host cell angiotensin-converting enzyme 2 receptor (ACE2), thus making the S protein the main target of neutralizing antibodies [39]. The large ectodomain of coronavirus S proteins includes two subunits, subunit 1 (S1), containing a receptor-binding domain (RBD), and the membrane-fusion subunit 2 (S2). The S protein is an ideal target for vaccine development on different platforms because it has a high antigenicity and the ability to induce a robust immune response [40]. Almost all types of COVID-19 vaccines run the endogenous synthesis of the SARS-CoV-2 spike protein. Synthetized S proteins move via blood circulation, interacting with ACE2 receptors and demonstrating the pathological features of SARS-CoV-2 [41]. A maximum concentration (14.6 μg/mL) of S proteins in blood serum was detected 24 h after vaccination and was reduced within 10 days [42]. Suggestions have been published that the spike protein may be responsible for the long-term effects of COVID-19, such as rare neurological complications, including Guillain–Barre syndrome and Bell’s palsy [43]. A growing body of research points to the potential dangers of the spike protein, even in the absence of the intact virus. The S protein has been reported to mediate pro-inflammatory and/or damaging (of various etiologies) responses in various human cell types [44,45]. Systemic inflammation, induced by the spike protein, may proceed through the TLR2-dependent activation of the nuclear factor kappa-light-chain-enhancer of activated B cells (NF-κB) pathway [46]. There are several in vitro studies suggesting the negative impact of the S protein on endothelial barrier function [47,48,49]. Other reports indicate that the S protein induces an inflammatory response in human corneal epithelial cells and binds to lipopolysaccharide (LPS), enhancing its pro-inflammatory activity [50,51]. Several publications have reported that the spike protein leads to hemagglutination, blood coagulation, and thrombosis [52,53]. Boschi et al., showed that the Wuhan, Alpha, Delta, and Omicron B.1.1.529 variants of the SARS-CoV-2 spike protein mixed with human erythrocytes led to hemagglutination [54].

There is evidence to suggest that alcohol consumption may cause the activation of the ACE2 receptor and, consequently, enhance the negative effect of the spike protein (Figure 1). Balasubramanian et al., observed an increase in ACE2 in brain expression in both chronic alcohol exposure and abrupt withdrawal from alcohol [55]. Reportedly, alcohol consumption induced the intracellular accumulation of reactive oxygen species (ROS), which leads to the activation of NF-κB and an increase in vascular endothelial growth factor (VEGF) and monocyte chemoattractant protein-1 (MCP-1) [56]. Moreover, alcohol could be an independent cause of syndromes, similar to COVID-19-vaccine-related side effects. A few recent clinical cases discussed the possibility that heavy alcohol consumption may play a role in the pathogenesis of Guillain–Barré syndrome [57,58]. Another study demonstrated that alcohol abuse was negatively associated with Bell’s palsy occurrence [59]. Thus, it can be assumed that alcohol consumption may provoke or enhance the “spike effect” of COVID-19 vaccines.

## 4. Protein Subunit Vaccines and Alcohol

Recombinant subunit vaccines contain purified and inactivated “subunits” of the pathogens. Their immunogenic properties can be amplified by immunopotentiating adjuvant systems or by means of targeting immunoreactive sites [60]. This approach of vaccine development has already been used for several other vaccines, including the recombinant hepatitis B vaccine, pneumococcal polysaccharide and meningococcal polysaccharide vaccines, pneumococcal conjugate and meningococcal conjugate vaccines, and recombinant influenza vaccine RIV4 [61].

The vaccine manufactured by Novavax (Gaithersburg, MD, USA) is the only recombinant subunit COVID-19 vaccine currently authorized for use in the United States by the FDA. The WHO added it to the Emergency Use Listing for 38 countries [62,63]. NVX-CoV2373 contains a saponin-based Matrix-M1 adjuvant and a recombinant SARS-CoV-2 nanoparticle vaccine from the full-length, wild-type SARS-CoV-2 S protein [61]. NVX-CoV2373 induces a relatively broad humoral and cellular immune response consisting of robust and polyfunctional CD4+ T cells and a modest CD8+ T cell response [64]. In Phase 3 clinical studies, most vaccine side effects were mild to moderate [62]. However, we can assume that since alcohol affects the function of T cells, the effectiveness of this vaccine would be weaker in people with alcohol-use disorders.

There are several ongoing clinical trials for other S-protein subunit recombinant COVID-19 vaccines based on the S1 subunit or RBD protein (Covax19, Nanocovax, SCTV01C, GBP510, etc.), showing good effectiveness and safety [61]. However, in our K18-hACE2 transgenic mice model, only subunit 1 of the SARS-CoV-2 S protein, instilled intratracheally, provoked acute lung injury and cytokine storm in lungs, unlike the whole S protein. We recently published a study demonstrating that K18-hACE2 transgenic mice on a Lieber–DeCarli ’82 ethanol liquid diet exhibit a more severe SARS-CoV-2 Spike Protein Subunit 1-induced acute respiratory distress syndrome (ARDS) than corresponding mice on a normal diet. Lung tissue homogenates from mice on alcohol diet showed the overexpression of ACE2 [21]. The S1 subunit of the SARS-CoV-2 spike protein exerted hippocampal neuronal cell death in mice, affecting brain functions [65]. More additional studies are needed to evaluate the safety of S-protein subunits on all organs and systems.

## 5. Inactivated Whole-Virus Vaccines and Alcohol

Vaccines based on inactivated pathogens have been used for over a hundred years as a protective agent against bacteria and viruses. Inactivated viral vaccines are first cultivated on a substrate (primary and continuous cell lines, tissues, fertilized eggs, and even whole organisms) to produce large amounts of antigens [66]. The multiplied virus in the substrate is purified, concentrated, and inactivated by various chemical agents (ascorbic acid, hydrogen peroxide, etc.) or by using physical methods (heat, ultraviolet exposure, gamma irradiation, etc.). In recent decades, only formaldehyde and β-Propiolactone have been used as inactivated agents for human viral vaccines [67]. Adjuvants are important components of many inactivated vaccines due to their ability to induce more robust and long-lasting specific immune responses [68]. Aluminum salts, such as aluminum hydroxide, phosphate, and potassium sulfate, have been widely used in vaccines for a long time [69].

Developed in China, the inactivated whole-virus vaccine Sinopharm (Beijing, China) BBIBP-CorV, containing an aluminum hydroxide adjuvant, has been approved by the WHO for emergency use, and it has been distributed in more than 40 countries [70]. Another Chinese vaccine approved by the WHO is CoronaVac (Sinovac (Beijing, China)), an inactivated SARS-CoV-2 aluminum-hydroxide-adjuvanted vaccine created from African green monkey kidney cells (Vero cells) that have been inoculated with SARS-CoV-2 [71,72]. In both BBIBP-CorV and CoronaVac clinical trials, alcohol addiction was one of the exclusion criteria [73]. No serious adverse reactions to vaccines, which could be aggravated by alcohol consumption, have been reported. A clinical study carried out by Jingwen Ai et al., demonstrated the safety of inactivated whole-virion SARS-CoV-2 vaccines in patients with alcoholic liver disease; however, those patients demonstrated a lower immunologic response to the vaccines than healthy patients [74].

## 6. Viral-Vector-Based COVID-19 Vaccines and Alcohol

In 1972, Jackson and colleagues created the recombinant DNA of the virus SV40, and in 1982, Moss used the vaccinia virus as a gene expression vector [75,76]. Vaccines based on viral vectors are able to intensify immunogenicity without an adjuvant, and they are able to induce a stable cytotoxic T-lymphocyte response in order to eliminate cells infected with the virus [77]. Vaccinia virus and adenovirus are the two most used vectors due to their abilities to induce a robust immune response against expressed foreign antigens and produce inflammatory cytokines and interferons [78]. This technology has recently proven itself in the production of Ebola vaccines and is now actively used for COVID-19 vaccines.

On 29 January 2020, the European Commission granted conditional marketing authorization for the Oxford/AstraZeneca COVID-19 vaccine (Covishield, Vaxzevria (Oxford, UK)), a monovalent vaccine composed of a single recombinant, replication-deficient chimpanzee adenovirus (ChAdOx1) vector encoding the S glycoprotein of SARS-CoV-2. However, five countries in the European Union have since placed age limitations on the vaccine, which has given rise to a certain distrust in it [79]. One of the rare but most severe side effects of this vaccine is a syndrome named vaccine-associated immune thrombosis and thrombocytopenia (VITT) [80]. Usually, the administration of a viral-vector-based COVID-19 vaccine induces the production of antibodies to the SARS-CoV-2 S protein. In very rare cases, VITT antibodies are generated that can bind to platelet factor 4 (PF4) and construct immune complexes that lead to a coagulation cascade and reduce the number of platelets [81]. As of April 2021, there had been 222 registered cases of VITT in Europe [80]. The Victorian Department of Health (Australia) equated this to eight cases of thrombopenia per million doses for the AstraZeneca vaccine [82]. Based on several reported cases, young women, especially those taking hormonal contraceptives, are at the highest risk of developing this vaccine-related adverse reaction [80]. Consequently, the AstraZeneca vaccine has not been authorized for use in the U.S. There have been no reports of Oxford/AstraZeneca-vaccine-related thrombosis and thrombocytopenia complications after alcohol consumption. However, binge alcohol consumption can lead to endothelial dysfunction, which, in combination with stasis and hypercoagulability, could increase venous thromboembolism (VTE) formation [83]. Liver dysfunction, caused by chronic alcohol intoxication, decreases the synthesis of anticoagulant thrombotic factors [84]. Nonetheless, there are also studies suggesting that low or moderate alcohol consumption could decrease the risk of deep venous thrombosis and pulmonary embolism in older people [85]. The ethanol treatment of human whole blood led to a decrease in PF4 release in response to a-thrombin [86]. According to Abolmaali’s study, AstraZeneca is the vaccine most reported to be associated with Guillain–Barré syndrome [87]. Summarizing the above facts, we can say that young people who drink alcohol, as well as those who chronically drink alcohol, have an increased risk of complications after immunization with the Oxford/AstraZeneca vaccine (Figure 2).

The next vaccine, approved by both the FDA and EUA for emergency use in February 2021, was the adenovirus-vector-based vaccine JNJ-78435735 developed by Johnson and Johnson (New Brunswick, NJ, USA) [57] along with Beth Israel Deaconess Medical Center (Boston, MA, USA) [88]. The clear advantage of this vaccine over other vector-based vaccines is that it is a single-shot vaccine. However, doctors faced a problem similar to that of the AstraZeneca vaccine—cases of a condition characterized by low platelets and thrombosis, including cerebral venous sinus thrombosis [89]. All cases of VITT occurred among women: 13 cases in 18–49-year-old women and 2 cases among women aged 50 years and older [90]. Following an emergency meeting that was held in December 2021, the use of mRNA COVID-19 vaccines was recommended over the Janssen COVID-19 vaccine [91]. The vaccine label information does not warn against alcohol use, but we can consider that alcohol abuse can increase the risk of VITT in young women.

The Gamaleya National Research Center for Epidemiology and Microbiology (Moscow, Russia) was the first to announce the creation of Gam-COVID-Vac (Sputnik V (Moscow, Russia)), a recombinant adenovirus-based vaccine [92]. Even though Sputnik V has not yet been approved by the WHO, it has been approved in 70 countries with a combined population of more than 4 billion people [93]. Sputnik V consists of two doses containing different components of the SARS-CoV-2 glycoprotein S gene, Ad26 and Ad5, administered separately 21 days apart [94]. The latter Ad’s immune complexes activate the dendritic T-cell axis [95]. A large proportion of this population, especially Africans, have high anti-Ad5 antibody titers from previous infections [96]. In an experiment on DO11.10 transgenic mice, it was shown that alcohol diminishes the capacity of dendritic cells to secrete interleukins IL-12 and IL-6 and reduces the ability to maintain the secretion of cytokines IL-17A and IFN-c but increases IL-13 expression [97]. Thompson et al., reported that ethanol promotes a reduced immune stimulatory capacity of female DC by reducing IL-12 production [98]. Thus, alcohol consumption after the second dose of the Sputnik V vaccine may significantly compromise its effectiveness, especially in some population categories. A Ministry of Health official representative warned that anyone being vaccinated against COVID-19 with Russia’s Sputnik V vaccine should give up alcohol for almost two months [99]. Interestingly, the Phase 3 trial on patients who received the Sputnik V vaccine showed only one patient with vein thrombosis unlike the above adenovirus-vector-based vaccines [93].

## 7. mRNA-Based Vaccines and Alcohol Consumption

The first report of protein production following reporter gene mRNA in mice was published by Wolff at al. in 1990 [100]. During that period, pharmaceutical companies did not consider mRNA a prospective technology because of doubts about its stability and its low efficacy [101]. Despite mRNA vaccines representing only 11% of all the developed COVID-19 vaccines, two mRNA vaccines, mRNA-1273 and BNT162b, were the first vaccines approved by the FDA and EUA for COVID-19 [102]. Both new mRNA vaccines, BNT162b2, manufactured by Pfizer/BioNTech, and mRNA-1273, produced by Moderna, contain molecules of RNA, modified with pseudo-uridine and encapsulated in a lipid nanoparticle vehicle. The Pfizer–BioNTech and Moderna vaccine constructs do not contain an S-protein S1/S2 furin cleavage site. Ribonucleic acid is endowed to be rapidly translated into nonactive SARS-CoV-2 S proteins in a stable closed structure in order to induce the immune response without causing cell damage due to its interaction with the ACE2 receptor [103]. However, these two vaccines were the most feared among people at the initial stage of vaccination due to the lack of data on their long-term side effects.

The S protein encoded by the vaccine is stabilized in its pre-fusion form; thus, it is possible that, if it enters the bloodstream and is distributed systemically throughout the human body, it may contribute to adverse effects [104]. Ndeupen et al., reported that the mRNA platform’s lipid nanoparticle (LNP) component used in preclinical vaccine studies causes a highly inflammatory response in mice. LNPs administrated intra-dermally, intramuscularly, or intranasally at a dose of 10 μg/mouse led to severe neutrophil infiltration, the activation of inflammatory pathways, and cytokine and chemokine production [105]. Such a reaction, in combination with the spike effect, can increase the negative consequences of vaccination in the body.

Among Japanese healthcare workers who were vaccinated with the BNT162b2 mRNA vaccine, alcohol consumption, along with other factors, was identified as a factor predicting lower IgG antibody titers after vaccination [106]. Wang et al., in their study of vaccinated patients with substance use disorders (SUDs), including alcohol disorders, demonstrated that patients with SUDs remain vulnerable to COVID-19 breakthrough infection, even after full vaccination. The risk was higher in patients who received the Pfizer-BioNTech vaccine than in those who received the Moderna vaccine [19].

Several cases of myocarditis have been reported following the administration of COVID-19 mRNA vaccines [107]. After the self-controlled case series, studies found that myocarditis after vaccination is higher in men younger than 40 years old, particularly after the second dose of the mRNA-1273 vaccine [108]. Excessive alcohol consumption can cause non-ischemic dilated cardiomyopathy and chronic heart disease, characterized by dilation and the impaired contraction of myocardial ventricles [109]. Of all alcohol-related myocardiopathy cases, 30% were myocarditis with a lymphocytic infiltrate in association with myocyte degeneration or focal necrosis [110]. Most people who heavily drink alcohol do not have any symptoms in the earlier stages of the disease, and many never develop clinical heart failure [111]. A case of vasospastic angina (VSA) caused by alcohol consumption following Pfizer/BioNTech vaccination has been reported [112]. Thus, a patient who chronically drinks alcohol, unaware of the presence of heart problems, could exacerbate them with an injection of the mRNA COVID-19 vaccine. Mark J. Mulligan et al., reported that up to 50% of patients demonstrated a decrease in lymphocytes after the first dose of the BNT162b1 vaccine [113], which, combined with the negative effect of alcohol on these cells, can have severe consequences for the immune system.

There is no data suggesting that other alcohol-associated chronic illnesses reduce the effectiveness of mRNA vaccines. Patients with compensated and decompensated cirrhosis demonstrated a 100% reduction in COVID-19-related hospitalization or death following the first dose of either the BNT162b2 or the mRNA-1273 vaccines [114].

## 8. Conclusions

To date, 24 COVID-19 vaccines have been approved by various institutions in different countries, with more than 100 vaccines undergoing clinical trials and more than 270 currently in pre-clinical development [115]. Besides the well-known adverse effects associated with antiviral vaccines, cases of severe pathologies and syndromes have been rarely observed among people who have received COVID-19 vaccines. Moreover, *The Lancet* reported that 1.3% of the cases processed by the Vaccine Adverse Event Reporting System (VAERS) in the USA were deaths [116]. Considering the risk of severe COVID-19 and the widespread distribution of vaccinations within a short time span, we can safely say that all these cases are insignificant compared to the benefits of vaccines. At the same time, the currently predominant Omicron strain variants have a reduced risk of severe disease, which, in turn, reduces the advantages of vaccination relative to the disadvantages. It was reported that the repeated use of vaccine boosters induced humoral and cellular tolerance against the Delta and Omicron variants [117]. There is no direct evidence in the literature indicating that moderate alcohol consumption has any effect on the health of vaccinated patients. However, there are several health conditions associated with alcohol abuse for which vaccination poses additional risks (Table 1).

At present, the “spike effect” of vaccines and its amplification by alcohol exposure is of most interest. More research is needed to understand the full mechanism of the alcohol-enhanced “spike effect” and to develop appropriate countermeasures to block it. It should also be considered that the chronic and excessive consumption of alcoholic beverages leads to a weakening of the immune system and, as a result, a lower effectiveness of vaccination.

## Figures and Tables

**Figure 1 pathogens-12-00163-f001:**
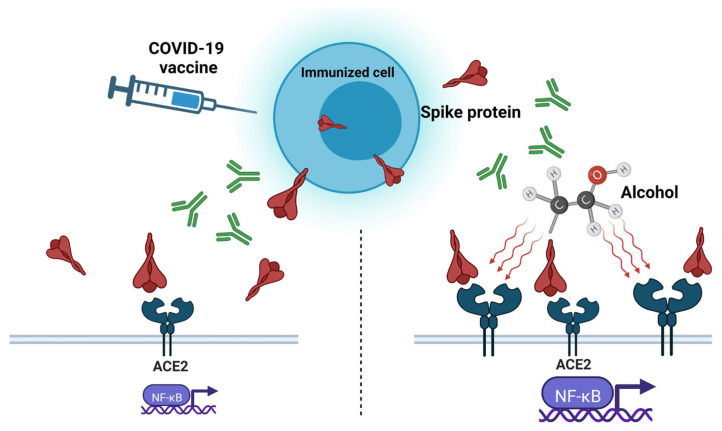
Ethanol-induced overexpression of angiotensin-converting enzyme 2 (ACE2) activates pro-inflammatory nuclear factor kappa-light-chain-enhancer of activated B cells (NF-kB) signaling pathway and exacerbates the “spike effect” of COVID-19 vaccines.

**Figure 2 pathogens-12-00163-f002:**
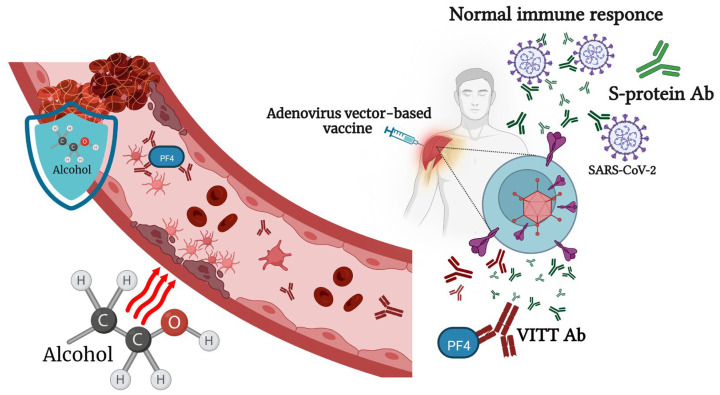
Alcohol consumption affects the immune response to adenovirus-vector-based vaccines and vaccine-associated immune thrombosis and thrombocytopenia (VITT). Physiologically relevant concentrations of alcohol lead to endothelial dysfunction, which, in combination with stasis and hypercoagulability, could increase venous thromboembolism (VTE) formation. At the same time, small or moderate doses of alcohol have an inhibitory effect on secondary platelet aggregation responses.

**Table 1 pathogens-12-00163-t001:** COVID-19 vaccines and post-vaccination risks associated with alcohol consumption.

Vaccine Name	Vaccine Type	Health Conditions and Potential Alcohol-Associated Risks	Reference
Sputnik V	Recombinant adenovirus	Low immune response (warning by health officials)	[99]
(Gamaleya National Research Centre for Epidemiology and Microbiology)		Possible reduced efficacy, especially in populations with high anti-Ad5 antibody titers from previous infections	[96]
JNJ-78435735	Recombinant adenovirus	Thrombosis and thrombocytopenia	[83,84,85,86,87]
(Johnson and Johnson/Beth Israel Deaconess Medical Center)			
Covishield, Vaxzevria	Recombinant adenovirus	Thrombosis and thrombocytopenia	[83,84,85,86,87]
(Oxford/AstraZeneca)			
BBIBP-CorV	Inactivated whole-virus vaccines	Low immune response	[74]
(Sinopharm)			
CoronaVac	Inactivated whole-virus vaccines	Low immune response	[74]
(Sinovac)			
NVX-CoV2373	Recombinant subunit	Low immune response	[22]
(Novavax)		High risk of “spike effect”	[21]
BNT162b2	mRNA	Cardiomyopathy	[109,110,111]
(Pfizer/BioNTech)		Low immune response	[106]
		Vasospastic angina	[112]
mRNA-1273	mRNA	Cardiomyopathy	[107,108,109]
(Moderna)		Low immune response	[106]

## Data Availability

Not applicable.

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
