# Peer review of "COVID-19 Vaccination and Alcohol Consumption: Justification of Risks"

_pathogens, 2023, doi:10.3390/pathogens12020163_

Round 1

Reviewer 1 Report

I was pleased to review the manuscript titled “COVID-19 Vaccination and Alcohol Consumption: Literature Justification of Risks”. The author presents a review that I consider to be very complete and highly relevant given the changes in alcohol consumption patterns during the COVID-19 pandemic. The manuscript will be of great interest to readers of Pathogens.

I only suggest doing a thorough review of the wording as there are some sentences that are a bit confusing (i.e. line 84: “Alcohol use does not have to be chronic to have negative health effects”).

Author Response

Dear Reviewer,
Thank you very much for your time involved in reviewing the manuscript and your very encouraging comments on the merits. As per your advice, some wording has been revised.

Reviewer 2 Report

Please confirm the attachment file.

Author Response

Dear Reviewer,
Thank you very much for your time spent reviewing the manuscript and your very valuable comments, which really helped improve the manuscript. All your advice and references have been taken into account and the conclusions have been revised.

Reviewer 3 Report

Your paper is correctly written and provides a significant scientific contribution

1.The main question that the researcher deals with is the influence of alcohol consumption on the effectiveness of vaccination against COVID 19 2.Due to the influence of alcohol on the effectiveness of vaccination and on health, it is certainly important to clarify this influence and to protect vaccine users in this way and encourage them to take such a preventive measure. Thus, the researcher was able to explain the unconscious approach of vaccine users towards alcohol consumption. 3. The conclusions are consistent with the evidence and arguments presented and they address the main question posent. 4.The researcher analyzes the data on the effect of alcohol on vaccination from the appropriate literature and thus forms a clear conclusion on the harmful effect of alcohol on vaccination. 5.The conclusions are consistent with the evidence and arguments presented and they address the main question pose. 6.The references are appropriate. 7.Tables and figures clearly show the content of the research

Author Response

Dear Reviewer,
Thank you very much for your time involved in reviewing the manuscript and your very encouraging comments.